# Bi- and Tri-Specific T Cell Engager-Armed Oncolytic Viruses: Next-Generation Cancer Immunotherapy

**DOI:** 10.3390/biomedicines8070204

**Published:** 2020-07-10

**Authors:** Zong Sheng Guo, Michael T. Lotze, Zhi Zhu, Walter J. Storkus, Xiao-Tong Song

**Affiliations:** 1UPMC Hillman Cancer Center, Pittsburgh, PA 15213, USA; lotzemt@upmc.edu (M.T.L.); zhuz4@upmc.edu (Z.Z.); storkuswj@upmc.edu (W.J.S.); 2Department of Surgery, University of Pittsburgh School of Medicine, Pittsburgh, PA 15213, USA; 3Department of Immunology, University of Pittsburgh School of Medicine, Pittsburgh, PA 15213, USA; 4Department of Dermatology, University of Pittsburgh School of Medicine, Pittsburgh, PA 15213, USA; 5Icell Kealex Therapeutics, Houston, TX 77021, USA; shautong.song@icellkealex.com

**Keywords:** oncolytic virus, bispecific antibody, trispecific antibody, bispecific T cell engager, trispecific T cell engager, tumor antigen-specific T cells, adenovirus, measles virus, vaccinia virus

## Abstract

Oncolytic viruses (OVs) are potent anti-cancer biologics with a bright future, having substantial evidence of efficacy in patients with cancer. Bi- and tri-specific antibodies targeting tumor antigens and capable of activating T cell receptor signaling have also shown great promise in cancer immunotherapy. In a cutting-edge strategy, investigators have incorporated the two independent anti-cancer modalities, transforming them into bi- or tri-specific T cell engager (BiTE or TriTE)-armed OVs for targeted immunotherapy. Since 2014, multiple research teams have studied this combinatorial strategy, and it showed substantial efficacy in various tumor models. Here, we first provide a brief overview of the current status of oncolytic virotherapy and the use of multi-specific antibodies for cancer immunotherapy. We then summarize progress on BiTE and TriTE antibodies as a novel class of cancer therapeutics in preclinical and clinical studies, followed by a discussion of BiTE- or TriTE-armed OVs for cancer therapy in translational models. In addition, T cell receptor mimics (TCRm) have been developed into BiTEs and are expected to greatly expand the application of BiTEs and BiTE-armed OVs for the effective targeting of intracellular tumor antigens. Future applications of such innovative combination strategies are emerging as precision cancer immunotherapies.

## 1. Introduction

Oncolytic viruses (OVs) are versatile and increasingly effective anticancer agents [1,2,3]. Four major mechanisms of action are engaged: oncolysis, vascular targeting, transgene expression and virus-elicited antitumor immune responses [3,4]. OVs preferentially infect and replicate in tumor cells and tumor-associated stromal cells [5,6]. When armed with extrinsic genes such as Th1-stimulatory cytokines, these virally expressed cytokines promote immune-mediated inflammatory responses and cytotoxic activities, leading to improved therapeutic outcomes. OVs modulate the tumor microenvironment (TME), resulting in augmentation of both local and systemic antitumor immunity. This is extremely important as systemic immunity is required for effective cancer immunotherapy in the setting of multi-focal, metastatic disease [7].

OVs represent an ideal platform to express therapeutic genes for multiple mechanisms of action [8,9], and can be utilized alone or in combination strategies [10,11,12]. Classes of *trans* genes that can be inserted into OV vectors include: (1) genes whose products can induce cancer cells to undergo apoptosis/necrosis, such as TRAIL [13,14]; (2) gene products that inhibit tumor-associated angiogenesis [15], such as IL-12 [16]; (3) Th1-stimulatory cytokines such as IL-2, IL-15 [17,18,19]; and (4) genes that encode antibodies that recognize one or more accessible tumor-associated and/or immune cell-associated antigens. Recombinant OVs can recondition the TME, facilitating entry, and they can sustain therapeutic functionality of tumor-infiltrating lymphocytes [19,20,21] in concert with antigen-crosspresenting dendritic cells and lymphatic vessel engagement [22,23], in association with improving antitumor efficacy. 

We have previously reviewed oncolytic immunotherapy in 2014 and 2017 [2,24]. Here, we focus on unique combinatorial OVs expressing antibodies that promote interaction between cancer/cancer-stromal cells with T or NK cells, enabling immune cell activation and tumoricidal activity. We will discuss the current status of the field of oncolytic virotherapy, integrating antibodies that are bi- or tri-specific into OVs for application in the cancer setting. Recent findings related to OVs armed with various BiTE antibodies for cancer immunotherapy will be reviewed.

## 2. Oncolytic Virus-Mediated Immunotherapy 

Viruses normally display three levels of tropism based on target cell species, tissue of origin and histologic lineage. Viral entry requires target cell surface expression of receptor(s) that determine cell permissiveness for viral infection and consequent outcomes, interacting with innate and adaptive immune components. However, when it comes to cancer cells, the species and tissue-type barriers restricting OV infection often disappear. This is most likely related to de-differentiation and metabolic reprogramming of cancer cells. The tumor selectivity of OV-infection has been well-studied [25,26]. OVs exert therapeutic activity via four distinct yet overlapping mechanisms: (1) oncolysis; (2) vascular targeting; (3) effector transgene expression; and (4) promotion of antitumor immunity [4]. Which of these four mechanisms is most important for treatment outcomes may vary depending on which OV was used as a therapeutic agent and what type of cancer cells are being treated. However, OV promotion of antitumor immunity consistently plays an important role in overall treatment efficacy. Immune stimulation occurs at many levels, including initial cross-priming of T cells, via a cascade mechanism involving tumor immunogenic cell death (ICD) induced by OV infection, replication and oncolysis, and subsequent presentation of danger signals to the dendritic cells that acquire, process and present tumor cell debris (containing tumor and viral antigens) to cognate T cells in tumor-draining lymph nodes or tumor-associated tertiary lymphoid structures [24,27,28]. 

OVs can coordinately activate both innate and adaptive immunity since they deliver PAMPs (pathogen-associated molecular pattern molecules) that initially activate innate immune cells and promote tumor immunogenic cell death (ICD), leading to the release of DAMPs (damage-associated molecular pattern molecules) as well as PAMPs that trigger DCs and their cross-presentation of tumor/viral antigens to T cells [24,27]. In fact, many investigators have designated OVs as therapeutic cancer vaccines [27,29,30,31,32]. As OVs can convert immune sparse (i.e., “cold”) tumors into immune-rich (i.e., “hot”) tumors [33,34], they appear to represent an ideal choice for combination with alternate therapeutic agents that require “hot” tumors for optimal biological efficacy. Rational combination of an OV with immune checkpoint blockade results in synergistic therapeutic effects in preclinical and clinical cancer models [20,33].

At least two attributes of OVs may dictate immune-mediated treatment outcomes. One of these is the inherent immunogenicity of the virus itself and its ability to promote tumor death, including ICD [35]. While most OVs induce ICD, some also eradicate tumor cells via alternate mechanisms with different intrinsic levels of associated immunogenicity. For example, when wild-type adenovirus (Ad), Semliki Forest virus (SFV) and vaccinia virus (VV) were examined for their abilities to induce tumor ICD, the infection of cancer cells by Ad was found to primarily activate tumor autophagy, necroptosis and pyroptosis, while SFV infection primarily activates ICD and VV primarily promotes necroptosis [36]. Immune reactivity to such dead/dying tumor cells may be further modulated by gene products expressed in armed OV, such as Th1-associated cytokines or other immunostimulants [37,38].

To date, only two OVs have been approved for clinical use in the world. Oncorine (H101) was approved for the treatment of nasopharyngeal carcinoma in combination with chemotherapy by Chinese authorities in 2005 [39], with more recent FDA approval of T-VEC (Imlygic™) for the treatment of advanced melanoma patients in the US in 2015 [40]. A major hurdle identified in these trials is that the TME is highly immunosuppressive, particularly in advanced stage patients, with single modality OVs incapable of effectively circumventing the immune dysfunctional TME. Recent attention has shifted to developing strategies using more potent OVs and combining OVs with other anti-cancer regimens that mitigate tumor-associated immune suppression while coordinately promoting therapeutic inflammatory responses [9].

Antitumor immunity elicited by OV is essential for, or at least contributes significantly to, the overall treatment efficacy of OV-mediated therapies. OV conditional promotion of immune-mediated cytotoxicity and anti-angiogenic activity within the TME also contributes to the treatment benefits associated with these therapeutic agents. Still, combination of OV with other immunotherapeutic modalities, such as cancer vaccines, immune checkpoint blockade, and multi-specific antibodies could diversify and reinforce synergistic anti-tumor mechanisms of action for multi-modality interventions, leading to improved treatment outcomes.

## 3. Bi- and Tri-Specific Antibodies, Bi- and Tri-specific T Cell Engagers (BiTEs and TriTE) for Integration into OV-Based Cancer Therapies 

Bispecific antibodies (bsAbs) recognize two individual epitopes, either on the same or on different antigens. The concept of bsAb was first described by Nisonoff and coworkers more than 60 years ago [41], with refinements from Segal and his team at the National Cancer Institute (NCI) [42]. Only in the last two decades have we seen rapid clinical evolution of bispecific antibodies and effective therapeutic applications [43]. Today, about 20 individual commercialized technology platforms are available for developing new bsAb. Two bsAbs have been marketed and about 80–90 are in various stages of clinical development [43,44,45]. These bi- and tri-specific (recognizing three distinct epitopes) antibodies possess significant potential for the effective therapy of solid tumor patients in future clinical trials [46].

By definition, the bispecific T cell engager antibody (BiTE) is a recombinant bispecific protein that has two linked single-chain fragment variables (scFvs) derived from two individual antibodies, one targeting a cell-surface molecule on T cells and the other targeting antigens on the surface of malignant cells. BiTEs are typically designed to bind to a selected tumor-associated antigen and to the invariant component of the T cell receptor (TCR) complex, i.e., CD3 chains that mediate activating signals into T cells (Figure 1). There have been numerous studies on this topic, with a PubMed search on “cancer, bispecific and antibody” identifying nearly 1000 publications. Here we will focus only on key recent reports. For older studies, please refer to excellent review articles for more details [44,47,48]. 

Trispecific antibodies binding to NK or T cells have also been explored to treat cancer (Figure 2). In one study, Vallera and colleagues designed IL-15 trispecific killer engagers (TriKE) based on their previous BiKE construct [50]. This TriKE contains a single-chain scFv against CD16 and CD33 to create an immunologic synapse between NK cells and CD33^+^ myeloid targets, as well as an IL-15 crosslinker that produces a trispecific engager to induce expansion, priming and survival of NK cells. When compared with the 1633 BiKE, the 161533 TriKE induced superior NK cell cytotoxicity, degranulation, and cytokine production against CD33^+^ HL-60 promyelocytic leukemia targets. In addition, the TriKE increased NK cell survival and proliferation. Specificity was demonstrated based on the selective ability of the 1615EpCAM TriKE to kill CD33-EpCAM^+^ target cells. In vivo, the 161533 TriKE exhibited superior antitumor activity and induced in vivo persistence and survival of human NK cells in an HL-60-*luc* tumor model for at least 3 weeks. 

CD28 costimulation provides another opportunity for therapeutic intervention, despite a checkered history in past therapeutic applications [51]. A trispecific antibody against CD3, CD28 and CD38 enhanced both T cell activation and tumor cell targeting [52]. The engagement of both CD3 and CD28 affords specific T cell activation, limits apoptosis/anergy, while provision of the anti-CD38 Ab recognizes myeloid cells as well as some lymphomas and leukemias. In a humanized mouse model, this trispecific T cell engager antibody (TriTE) treatment suppressed myeloma growth. It also stimulated memory/effector T cell proliferation and reduced T*reg* cell levels in non-human primates. Collectively, these studies suggest that trispecific antibodies represent a promising platform for cancer immunotherapy.

Several completed or ongoing clinical trials have used/are using BiTE or bsAb for treatment of patients with various malignancies, including those with solid cancers [46,54]. Blinatumomab is a CD19/CD3 bsAb antibody designed for the treatment of a number of blood cancers. Single-agent blinatumomab showed anti-lymphoma activity [55]. Blinatumomab monotherapy appears effective in patients with relapsed/refractory diffuse large B-cell lymphoma, a heavily pretreated patient population with a high unmet medical need [56]. In a multi-institutional phase 3 trial, adults with heavily pretreated B-cell precursor acute lymphoblastic leukemia (B-ALL) were randomly assigned to receive either blinatumomab or standard-of-care chemotherapy. The primary end point was overall survival. The results showed that overall survival was significantly longer in the blinatumomab treatment group vs. the chemotherapy treatment group. The median overall survival was 7.7 months in the blinatumomab treatment group and 4.0 months in the chemotherapy treatment group [57]. Children’s Oncology Group Study AALL1331 reported last year that in a randomized phase 3 trial, blinatumomab as post-reinduction therapy in high and intermediate risk cases of first relapse of B-ALL in children and adolescents/young adults also demonstrated superior efficacy and greater tolerability than chemotherapy [58]. After a number of clinical studies, blinatumomab (Blincyto) was first approved by the FDA in 2014 for treatment of patients with B-ALL; in 2017 it received full approval to treat relapsed or refractory B-cell precursor ALL in adults and children; and in 2018, it expanded to the treatment of patients with B-cell precursor acute lymphoblastic leukemia (ALL) in morphologic remission with minimal residual disease [59]. 

A second bsAb, catumaxomab, is a rat–mouse hybrid monoclonal antibody used to treat malignant ascites, a condition occurring in some patients with metastatic disease. It coordinately binds to the antigens CD3 (T cells) and EpCAM (carcinoma). It was approved in Europe on 20 April 2009 for the treatment of malignant ascites in patients with EpCAM-positive cancer if a standard therapy is not available [48]. However, it was voluntarily withdrawn later from the US market and the EU market for commercial reasons.

AMG420, the anti-B-cell maturation antigen BiTE molecule, was assessed in patients with relapsed/refractory multiple myeloma [60]. In this first-in-human study, clinicians administered up to 10 cycles of AMG420 (4-week infusions/6-week cycles) to patients. The response rate was 70%, including 50% minimal residual disease (MRD)-negative complete responses at 400 μg/d, the maximum tolerated dose (MTD) for this study. These results suggested that AMG402 is a highly promising drug for patients with relapsed/refractory myeloma.

Some BiTEs are being tested in solid tumor models, including BiTE constructs against CD3/HER2. One study found that their BiTE selectively targets HER2-overexpressing cancer cells with high potency, while sparing normal cells that express low levels of HER2 [61]. In another study, relative target affinities of the BiTE for either CD3 or HER2 determined biodistribution in a solid tumor mouse model [62], suggesting that additional protein engineering might be required. Additional BiTEs have been evaluated in human patients with refractory solid tumors [45,63]. One example is a BiTE that targets CD3 and EpCAM, termed solitomab (MT1110, AMG110). Patients with relapsed/refractory EpCAM-positive solid tumors treated with solitomab developed dose-limiting toxicities (DLTs), including severe diarrhea and increased liver enzymes, which precluded dose escalation to potentially therapeutic levels. Since systemic application of BiTE may lead to severe toxicities, integration of such agents into alternate vehicles with tumor tropism (such as OVs) may coordinately reduce systemic toxicity and increase local delivery of BiTEs to therapeutic levels within the TME. 

There remain significant drawbacks associated with therapeutic application of BiTE/TriTE or bsAb/tsAb molecules. These molecules typically exhibit short biologic half-lives, rapid blood clearance, fast off-rates, and poor retention times in targeted sites (e.g., tumors) [46]. A second disadvantage is that despite their ability to turn solid tumors into inflammatory sites, they do not promote durable protective memory [64]. In contrast, OVs provide long-term protective anti-tumor immunity, particularly when combined with cancer vaccines [65]. Furthermore, the efficacy of BiTE and TriTE is regulated through diverse checkpoint molecules [66], suggesting obvious synergy with immune checkpoint inhibitors to achieve the optimal therapeutic benefit.

In contrast, the major advantage for BiTE and TriTE molecules is that they provide “specificity” for polyclonally-activated populations of T cells, making them refractory to tumor immune evasion mechanisms, including loss of MHC molecule expression [67]. In summary, despite some obvious disadvantages, there remains significant potential for the further development of this class of antigen-specific and immune cell-specific orchestrating molecules as cancer immunotherapeutic agents, especially when used in combination with OVs to overcome their operational limitations [45,54,68]. 

## 4. BiTE- and TriTE-Armed OVs Mediate Superior Therapeutic Efficacy

Our group (XT Song) hypothesized that arming OV with a gene encoding a bispecific antibody coordinately engaging tumor cells and T cells would improve the antitumor activity of oncolytic VV (Figure 3). An oncolytic VV encoding a secretory BiTE consisting of two single-chain variable fragments, specific for CD3 and the tumor cell surface antigen EphA2 (EphA2-T cell engager), was constructed and designated as EphA2-TEA-VV [69] (Table 1). This BiTE had been previously shown to possess selective targeting and potently controlled tumor growth [70]. In this study, it was demonstrated that the virus infected and replicated within and induced oncolysis of tumor cells in vitro, to a degree similar to that of the un-armed virus. Indeed, tumor cells infected with EphA2-TEA-VV induced T cell activation, as evidenced by interferon-γ and interleukin-2 secretion. In coculture assays, this armed OV not only killed infected tumor cells, it also induced bystander killing of noninfected tumor cells in the presence of T cells. Subsequent in vivo studies in a lung cancer xenograft model expressing the tumor antigen EphA2 revealed that EphA2-TEA-VV, when applied in combination with adoptively transferred human T cells, mediated superior antitumor activity when compared with control VV plus T cells. Therefore, BiTE-armed OV represents a promising approach to improve oncolytic immunotherapy [69].

Oncolytic adenoviruses (AdV) expressing various BiTEs have been evaluated by several research teams. Fajardo et al. tested the hypothesis that tumor-infiltrating lymphocytes (TIL) could be more effectively activated and redirected by an OV armed with BiTE antibodies [72]. The oncolytic AdV was engineered to express an EGFR-targeting BiTE antibody, which was designated as ICOVIR-15K-cBiTE. This virus retained its oncolytic properties in vitro. Furthermore, BiTE expression and secretion were detected in supernatants from the virus-infected cells, with secreted BiTEs specifically binding both CD3^+^ and EGFR^+^ expressing cells. In cell cocultures, ICOVIR-15K-cBiTE-mediated oncolysis and cBiTE expression resulted in robust T cell activation and proliferation, and additionally supported bystander cell-mediated cytotoxicity. In two distinct tumor xenograft models, the combined delivery of the armed OV with peripheral blood mononuclear cells (PBMCs) or T cells enhanced the antitumor efficacy achieved by either component as a monotherapy. An additional recent study showed that an EGFR-targeted BiTE-armed OV could be delivered into the TME using mesenchymal stem cells as carriers for improved efficacy [79].

The oncolytic group B adenovirus EnAdenotucirev (EnAdV) was modified by Freedman and colleagues to express another BiTE coordinately binding EpCAM^+^ tumor cells and CD3^+^ T cells, leading to clustering and activation of both CD4^+^ and CD8^+^ T cells with tumor cells [73]. Using the same logic, cancer cells infected with oncolytic measles virus-encoding BiTEs (MV-BiTEs) secreted functional BiTE antibodies. More importantly, they demonstrated therapeutic efficacy of MV-BiTE against established tumors in fully immunocompetent mice. In this model, antitumor efficacy was associated with increased infiltration of TIL with the induction of durable protective antitumor immunity. The therapeutic efficacy of MV-BiTE in xenograft spheroid models of patient-derived primary colorectal carcinoma could be demonstrated if co-administered with human PBMCs [74].

Fibroblast activation protein-α (FAP) is overexpressed in cancer-associated fibroblasts (CAFs), the primary constituent of tumor stroma. As a consequence, several investigators have used FAP as a target for BiTE development [80]. We (X-T Song) constructed an oncolytic VV encoding a BiTE specific for murine CD3 and fibroblast activation protein (mFAP-TEA-VV). In vivo mFAP-TEA-VV replicated within tumor sites and mediated potent antitumor activity when compared with control VVs in an immunocompetent B16 mouse melanoma model. Remarkably, the improved viral spread of mFAP-TEA-VV correlated with the destruction of tumor stroma [75]. Based on this same principle, the Seymour group generated an OV expressing a BiTE that simultaneously targets cancer and immunosuppressive stromal cells was generated [76]. This BiTE binds FAP on CAFs and CD3ε on T cells, leading to fibroblast cell death and potent T cell activation. Indeed, treatment of fresh clinical biopsies, including malignant ascites and solid prostate cancer tissue, with FAP-BiTE-encoding OV induced expression of PD-1^+^ on TIL with subsequent lysis of CAFs. Another BiTE-armed AdV co-targeting FAP used an anti-human CD3 scFv linked to an anti-murine and human FAP scFv. This FBiTE was inserted in the oncolytic adenovirus ICOVIR15K under the control of the major late promoter, generating the ICO15K-FBiTE. In vivo, T cell biodistribution and antitumor efficacy were evaluated. FBiTE binding to CD3^+^ effector T cells and FAP^+^ target cells led to T cell activation, proliferation, and the cytotoxic death of FAP-positive cells. In Hu-SCID tumor models, FBiTE expression in OV enhanced intratumoral accumulation of T cells and decreased the level of FAP expression in treated tumors. The antitumor activity of the FBiTE-armed OV was superior to the parental virus [77]. Collectively, all of these studies suggest that this type of BiTE-armed OVs can target both cancer cells and the tumor-associated stroma to promote enhanced therapeutic efficacy.

Certainly, no single cancer immunotherapy will likely be effective in patients with solid tumors due to a variety of mechanisms of resistance to immunotherapy [81]. These include heterogeneity of tumor antigen expression in individual tumor primary sites and metastases, and active immune suppression by the tumor environment [82]. Several unique and shared major hurdles are associated with personalized therapeutic strategies [83]. Tumor antigen heterogeneity is a major issue for both therapies using CAR T cells or BiTE-OVs. To overcome this major issue, two studies have been performed with BiTE-OVs in combination with adoptively transferred CAR T cells. Wing and colleagues showed that CAR T cells targeting folate receptor alpha (FR-α) successfully infiltrated xenografted tumors, but failed to achieve complete responses, presumably due the presence of FR-α-negative cancer cells induced by tumor escape. Treatment efficacy of a combination of FR-α-targeted CAR T cells was subsequently combined with an EGFR-targeting BiTE-expressing oncolytic adenovirus (Ad), ICO15K-cBiTE [84]. The Ad-BiTE-mediated oncolysis enhances activation and proliferation of CAR T cells, even in the absence of expression of FR-α, and improved tumor killing mediated by CAR T cells. The combination treatment improved T cell activation and antitumor efficacy, and prolonged survival in cancer models in mice. Suzuki and collaborators used an OV that simultaneously produces IL-12, an anti-PD-L1 antibody, and a BiTE molecule (forming CAdTrio). CD44v6 BiTE expressed from CAdTrio enabled HER2-specific CAR T cells to kill multiple CD44v6^+^ cancer cell lines and to produce more rapid and sustained disease control of orthotopic HER2^+^ and HER2^−^ CD44v6^+^ tumors. CAdTrio, when used in combination with HER2.CAR T cells, ensured dual targeting of two tumor antigens by engaging distinct classes of the receptor (CAR and native receptor [TCR]), and improved therapeutic outcomes [85]. 

In 2019, Scott et al. developed both BiTE-and TriTE-armed Ad viruses and showed that they can be used to deplete tumor-associated macrophages in cancer patient samples [78]. BiTEs/TriTEs were designed to recognize CD3ε on T cells, along with CD206 or folate receptor β on M2-like macrophages. This is the first study to achieve selective depletion of specific M2-like macrophage subsets, opening the door for in vivo testing the ability of these agents to eradicate cancer-supporting tumor-associated macrophages (TAMs). 

The success of BiTE- and TriTE-based cancer therapy depends on the identification of antigens expressed differentially in tumors and less on normal tissues. The available tumor antigens can be classified into five groups, namely, tissue differentiation antigens, tumor germline (“tumor-testis”) antigens, normal proteins overexpressed by cancer cells, viral proteins and tumor-specific mutated antigens (neoantigens) [86,87]. The choice of the best antigens for targeting is made more difficult given a hallmark characteristic of most forms of cancer, antigenic heterogeneity [88,89], which can differ between regions in a given lesion in the patient and between lesions in patients with multi-focal disease [86,90,91]. Therefore, preferred iterations of BiTE/TriTE-armed OV-based therapies represent forms of cancer precision medicine [92,93,94], requiring deep insights into the disordered tumor microenvironment and the precise mechanisms/agents to overcome them.

## 5. Despite Many Strengths, There Remain Significant Hurdles for This Approach as an Effective Immunotherapy

There are several major advantages in applying BiTE and TriTE in the cancer setting. One is that a therapeutic BiTE/TriTE can target intracellular proteins in tumor cells, such as oncoprotein WT1 [95]. T cell receptor mimics (TCRm) or T cell receptor (TCR)-like antibodies recognize “aggretopes” on the peptide/MHC-I bimolecular complex, and this recognition is similar to that mediated by T cell expressed TCR [96,97]. These antibodies could be further developed as BiTEs [95,98]. This property greatly expands the utility of this approach as there are fewer antigens expressed on the tumor cell surface when compared with the intracellular compartment. Furthermore, polyclonally BiTE-activated T cells recognize tumor cells regardless of expression of MHC class I [67]. In addition, OV delivering BiTE/TriTE directly to tumors is expected to reduce biologic dosing, thereby limiting the commonly observed systemic side effects of BiTE molecules. 

As a promising approach to treatment of solid tumor patients, BiTE- and TriTE-based drugs still face enormous challenges for application against common cancers, given observed tumor heterogeneity and variable mutational burden, poor drug delivery/penetration into tumors, an overwhelmingly immunosuppressive TME, severe systemic side effects, and off-target toxicities against adjacent normal cells [99,100]. 

As for OVs, there are many strengths, including converting immunologically cold tumors into hot ones [33,34], a prerequisite for predicting beneficial response to standard-of-care immunotherapies, such as checkpoint inhibitors. In addition, the induction of long-term protective immunity against tumors appears to require concerted recognition of tumor antigens by memory CD4^+^ and CD8^+^ T cells, and this may be best achieved by treatment with cancer vaccines plus OV [65]. Regardless, many challenges lie ahead for OVs’ clinical evolution as immunotherapies, which have been discussed in recent reviews by a number of teams, including our own [2,26,101,102]. They may include, but are not limited to (1) tumor antigenic heterogeneity; (2) insufficient delivery of OVs into the TME; (3) the generation of neutralizing antiviral immunity following exposure to the virus, limiting the effectiveness of subsequent treatments with the same virus; and (4) immune escape, evasion and suppression in the TME.

BiTE- and TriTE-armed OVs, which merge two complementary treatment strategies into one vehicle, provide an exciting means to restrict production of BiTEs/TriTEs within tumor sites, while widening the therapeutic window, permitting synergy with additional complementary interventional strategies. Localizing replication of the OV and thus expression of BiTE/TriTEs to tumor tissues should overcome key pharmacokinetic issues associated with their systemic delivery, as well as avoiding on-target off-tumor toxicities of these BiTE/TriTE [99]. Thus far, approximately ten independent studies provide reason for excitement in the translational development of these agents as monotherapies or combination immunotherapies with other novel therapeutic agents, such as CAR T cells or TIL. 

## 6. Conclusions and Perspectives

The field of cancer immunotherapy launched in the mid-1980s with the availability and subsequent approval of interleukin 2 therapy for patients with melanoma and renal cancer. Since 2010, several approaches from bench-to-bedside, including immune checkpoint blockade, genetically engineered T and NK cells (CAR T, CAR NK, and TCR T cells), tumor-infiltrating lymphocytes (TIL) and other immune cells, neoantigen-targeted cancer vaccines, and combination strategies, including one or more of these approaches, have been developed. Oncolytic immunotherapy, likewise, is a fast-moving field of therapeutics development. Armed OVs, including T-VEC, have been approved by the FDA and EU authorities for the treatment of patients with advanced-stage melanoma [40], serving as a model for OVs in development. As for BiTEs, blinatumomab, a dual-specific antibody against CD19 and CD3, has yielded improved outcomes for patients with B cell lymphoma [103], or B-cell precursor acute lymphoblastic leukemia in morphologic remission with minimal residual disease [59]. 

Despite significant progress, most cancer patients with advanced-stage solid tumors remain unresponsive to immunotherapy, or they develop acquired resistance to such interventions. Investigators have been testing various strategies to overcome the resistance to immunotherapy [81]. We believe that the application of BiTE- or TriTE-armed OVs represents a powerful and effective strategy to address this unmet clinical need, particularly when combined with complementary approaches designed to mitigate the immunosuppressive TME [104,105]. Such efforts should lead to the development of novel anti-cancer agents including superior T cell engagers [106,107], which represents a top-10 challenging task in the field of cancer immunotherapy [108]. It is clear that more powerful OVs will be developed in the process [38]. In addition, navigating metabolic pathways to enhance and sustain antitumor immunity and the benefits of immunotherapy is a key consideration [109,110]. Ultimately, one may need to combine three or more components targeting multiple core pathways in the TME using T cell engager–OV, small molecule drugs targeting key components in the TME, plus immune checkpoint inhibitors [66,111] to enable objective clinical responses and define the next-generation standard of care for cancer patients. 

## Figures and Tables

**Figure 1 biomedicines-08-00204-f001:**
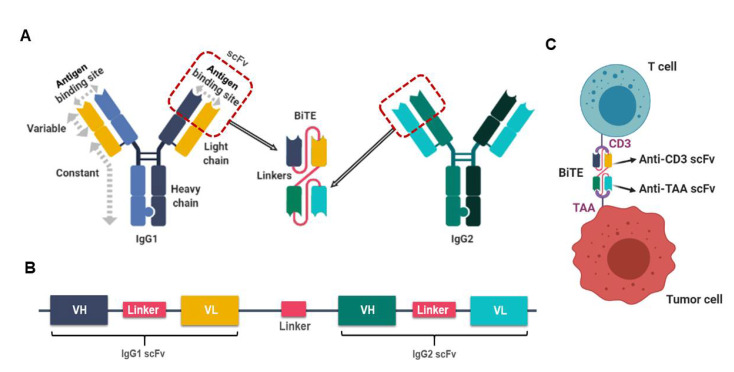
The design of a bispecific T cell engager antibody (BiTE). (**A**). Schematic representation of the derivation and structure of a BiTE molecule generated from two different antibodies, one with specificity for a T cell activation molecule and another one specific for a tumor-associated antigen (TAA). (**B**). Schema of a BiTE-coding gene used to produce the recombinant BiTE protein. Linkers were inserted between VH and VL domains of the single-chain fragment variable (scFv) and between the two scFvs. The linker between the two scFv is short (~5 aa) while the other linker within scFv is usually longer (~15 aa). (**C**). A BiTE creates an immunologic synapse by binding simultaneously to a tumor cell, via TAA, and a T cell, via CD3. This figure is modified from the Figure 1 in Slaney CY et al., Cancer Discovery, 2018 [49]. The drawings were created using BioRender (https://app.biorender.com).

**Figure 2 biomedicines-08-00204-f002:**
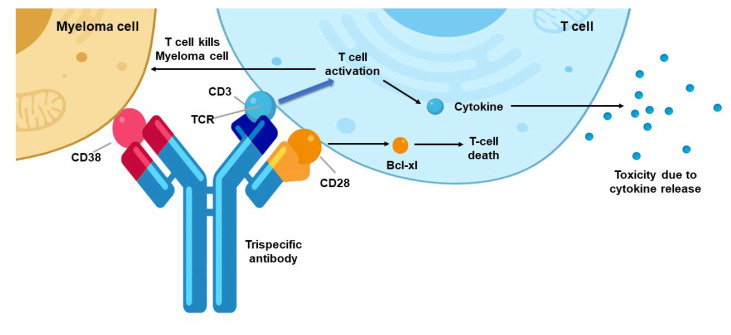
The design of a triple specific T cell engager (TriTE) antibody and how it links the T cells to the targeted cancer cells. This trispecific antibody binds three targets: the CD38 protein on a myeloma cell, and the protein CD28 and the CD3 protein complex on a T cell (the antibody’s target-binding domains are shown in red, blue and yellow, respectively). CD3 is a component of the T cell receptor (TCR). The binding of CD3 by the antibody drives T cell activation without requiring antigen recognition by the TCR, which leads to the killing of the myeloma cell and the production and release of toxic cytokine molecules. This image is modified from that in Garfall AL and June C, Nature, 2019 [53], and it was made using BioRender.

**Figure 3 biomedicines-08-00204-f003:**
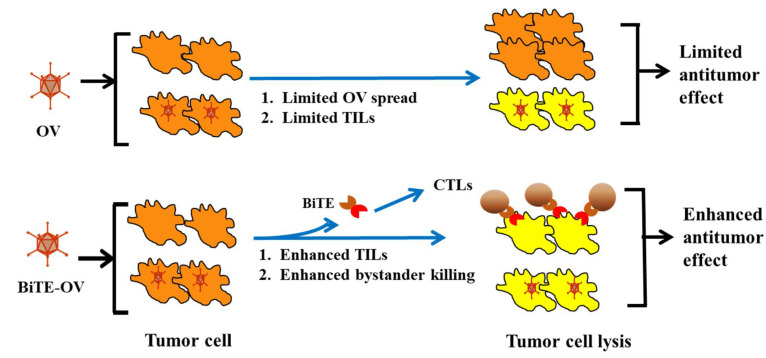
Working mechanisms of a BiTE-armed oncolytic virus (OV). The efficacy of a “pure” OV is often limited by suboptimal spread of the OV throughout the tumor tissue and induction of tumor antigen-specific T cells. BiTE-armed OV may overcome these limitations as the armed OV produces and secrete these BiTEs that diffuse within the tumor tissue, activating and directing endogenous T cells to recognize and kill the tumor cells or/and stromal cells effectively (even if not directly infected by the OV), resulting in improved antitumor efficacy. This is a modified version of a figure originally published by Song XT, Discovery Med, 2013 [71].

**Table 1 biomedicines-08-00204-t001:** Representative BiTE- and TriTE-armed OVs for cancer immunotherapy.

OV	BiTE or TriTE	Tumor Model	Main Observations	Reference
VV	CD3/EphA2	Human lung cancer in SCID/ beige mice	(1). Tumor cells infected with the OV-induced T cell activation. In coculture assays, the armed OV not only killed infected tumor cells, but in the presence of T cells, it also induced bystander killing of non-infected tumor cells. (2). EphA2-TEA-VV combined with human T cells had potent antitumor activity.	[69]
Ad	CD3/EGFR	Human lung and colorectal cancers in SCID/beige mice	(1). ICOVIR-15K-cBiTE–mediated oncolysis resulted in robust T cell activation, proliferation, and bystander cell-mediated cytotoxicity. (2). Intratumoral injection increased the persistence and accumulation of TILs in vivo, and combined delivery with PBMCs or T cells enhanced antitumor efficacy.	[72]
EnAd	CD3/EpCAM	Primary pleural effusions and peritoneal malignant ascites	Infection of cancer cells leads to the activation of endogenous T cells to kill endogenous tumor cells.	[73]
Measles viruses	CD3/CEA or CD20	(1). MC38-CEA model; (2). Human primary colorectal cancer-derived spheroids	(1). Therapeutic efficacy against established tumors in fully immunocompetent mice. (2). Therapeutic efficacy in xenograft models of patient-derived primary colorectal carcinoma spheroids with transfer of PBMCs.	[74]
VV or Ad	CD3/FAP	(1). B16 models; (2). Human colon and lung cancers	BiTE-armed OVs combine direct oncolysis of cancer cells with endogenous T cell activation to attack stromal fibroblasts.	[75,76,77]
EnAd	CD3, CD206 folate receptor β	Human cancer samples in vitro	T cells activated by BiTE- or TriTE-armed EnAd preferentially killed M2-like autologous macrophages.	[78]

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
