# Peer review of "Bi- and Tri-Specific T Cell Engager-Armed Oncolytic Viruses: Next-Generation Cancer Immunotherapy"

_biomedicines, 2020, doi:10.3390/biomedicines8070204_

Round 1
Reviewer 1 Report
The review was well written and provided a good summary of the literature.
Some minor typographical errors were detected.
- line 234 - the type of interferon was missing.
- Sentence starting on line252 required attention.
- line 272 - delete "a" to fit with the plural version of the sentence?
- line 293 - change to "this type of BITE"
- line 297 - should read sites
- line 339 - check you have defined SOC somewhere and use throughout after first abbreviation.
- line 340 - space missing between checkpoint inhibitor
- line 360 - delete comma
Author Response
Thank you for your insightful comments and suggestions.
You pointed out that there are 8 minor typographic errors. These have all now been corrected as suggested in the revised version.
Reviewer 2 Report
The manuscript submitted by Guo and colleagues provides a recent review of literature pertaining to the development and use of BiTE- and TriTE-armed oncolytic viruses. Considerations of their strengths and limitations are also briefly discussed. All in the all, this review article is well-researched, timely, and should be of interest to those performing research in the related fields and immunotherapy in general. I recommend the article for publication following some minor revisions and editing:
- Some acronyms are used without being properly written out first whereas others are written out and abbreviated multiple times. Examples include DLBCL (line 173), ALL (lines 175, 185, 187), and OVV (lines 228 and 276). Likewise, Ad, AdVs, and oAd are used at various points in reference to oncolytic adenoviruses; one abbreviation should be chosen and used for consistency.
- Line 301-302 “Guedan and June teamed up and showed…” sounds too informal and should be replaced with something like “Wing and colleagues showed…” in reference to the first author of the cited paper. In the same section (line 308), “In vivo, the combination treatment improves T cell activation and antitumor benefits” is awkwardly phrased and should be re-worded for clarity.
- Lines 309-310 “an OV that simultaneously produces IL-12, a checkpoint blockade molecule, and a BiTE molecule” should specify what checkpoint blockade molecule is encoded by the OV.
- Lines 338-340 “As for OVs, there are many strengths including, (1) converting immunologically cold tumors into hot ones [33,34], a prerequisite for predicting beneficial response to SOC immunotherapies, such as checkpointinhibitors” is structured as if it is going to provide a list of positive attributes, but ends there. The sentence should either be modified or the (1) should be removed.
- Section 6, BiTE/TriTE and their armed OVs as cancer precision medicines, currently feels somewhat misplaced in the review and might be better suited as a conclusion of sorts to section 4.
Author Response
Concerning the 5 minor suggested revisions and editing issues.
- About acronyms. You are absolutely correct. We have now defined those that were not previously spelled out, while eliminate others. (1). We now spell out DLBCL, i.e. diffuse large B-cell lymphoma. (2). “OVV” has been changed to oncolytic VV. (3). The acronym “ALL” has been spelled out as “acute lymphoblastic leukemia” when it first appears. (4). The B-cell precursor acute lymphoblastic leukemia has been shortened to B-ALL (Line 177). (5). The acronyms for oncolytic adenovirus (Ad, AdVs, and oAd) have been all been converted to “AdV” for internal consistency.
- Lines 301-302 (Lines 309-310 in the revised version): We changed to “Wing and colleagues”, accordingly. Line 308 (Lines 34-325 in the revised version): we have re-worded the sentence for clarity.
- Lines 309-310 (Lines 317-318 in the revised revision): We now specify that it is anti-PD-L1 antibody (in addition to IL-12 and a BiTE) that is encoded in the OV genome.
- Lines 338-340 (Lines 361-362 in the revised version): We have made the suggested change.
- Section 6. Based on your suggestion, we have moved the paragraph to section 4, thereby eliminating old section 6.
Thank you again for your insightful comments and suggestions.